# Dupilumab Efficacy on Asthma Functional, Inflammatory, and Patient-Reported Outcomes across Different Disease Phenotypes and Severity: A Real-Life Perspective

**DOI:** 10.3390/biomedicines12020390

**Published:** 2024-02-08

**Authors:** Marco Caminati, Matteo Maule, Roberto Benoni, Diego Bagnasco, Bianca Beghè, Fulvio Braido, Luisa Brussino, Paolo Cameli, Maria Giulia Candeliere, Giovanna Elisiana Carpagnano, Giulia Costanzo, Claudia Crimi, Mariella D’Amato, Stefano Del Giacco, Gabriella Guarnieri, Mona-Rita Yacoub, Claudio Micheletto, Stefania Nicola, Bianca Olivieri, Laura Pini, Michele Schiappoli, Rachele Vaia, Andrea Vianello, Dina Visca, Antonio Spanevello, Gianenrico Senna

**Affiliations:** 1Department of Medicine, University of Verona, Piazzale L.A. Scuro, 37134 Verona, Italyrachele.vaia@univr.it (R.V.); gianenrico.senna@univr.it (G.S.); 2Allergy Unit and Asthma Center, Verona Integrated University Hospital, 37126 Verona, Italy; biancaolivieri92@gmail.com (B.O.); michele.schiappoli@aovr.veneto.it (M.S.); 3Department of Diagnostics and Public Health, University of Verona, 37129 Verona, Italy; roberto.benoni90@gmail.com; 4Allergy and Respiratory Diseases Clinic, IRCCS Policlinico San Martino, University of Genoa, 16132 Genoa, Italy; diego.bagnasco@dimi.unige.it (D.B.); fulvio.braido@unige.it (F.B.); mariagiulia.candeliere@unige.it (M.G.C.); 5AOU of Modena, University of Modena and Reggio Emilia, 41125 Modena, Italy; bianca.beghe@unimore.it; 6SCDU Immunologia e Allergologia, AO Ordine Mauriziano di Torino, 10137 Torino, Italy; luisa.brussino@unito.it (L.B.); stefania.nicola@unito.it (S.N.); 7Respiratory Diseases and Lung Transplantation Unit, Department of Medicine, Surgery and Neuroscience, University of Siena, 53100 Siena, Italy; paolo.cameli@unisi.it; 8Institute of Respiratory Disease, Department Translational Biomedicine and Neuroscience, University “Aldo Moro”, 70121 Bari, Italy; elisiana.carpagnano@uniba.it; 9Department of Medical Sciences and Public Health, University of Cagliari, 09124 Cagliari, Italy; giuliacostanzo14@gmail.com (G.C.); stedg@medicina.unica.it (S.D.G.); 10Department of Clinical and Experimental Medicine, University of Catania, 95124 Catania, Italy; dott.claudiacrimi@gmail.com; 11Respiratory Medicine Unit, AOU Policlinico “G. Rodolico—San Marco”, 95123 Catania, Italy; 12Respiratory Department, Monaldi Hospital AO Dei Colli, Federico II University, 80131 Naples, Italy; marielladam@hotmail.it (M.D.); andrea.vianello1@unipd.it (A.V.); 13Department of Cardiac, Thoracic, Vascular Sciences and Public Health, University of Padova, 35122 Padova, Italy; gabriella.guarnieri@unipd.it; 14Unit of Immunology, Rheumatology, Allergy and Rare Diseases, IRCCS San Raffaele Scientific Institute, 20132 Milan, Italy; yacoub.monarita@hsr.it; 15Pulmonology Unit, Verona Integrated University Hospital, 37126 Verona, Italy; claudio.micheletto@aovr.veneto.it; 16Respiratory Medicine Unit, ASST—“Spedali Civili” of Brescia, 25123 Brescia, Italy; laura.pini@unibs.it; 17Division of Pulmonary Rehabilitation, Istituti Clinici Scientifici Maugeri, IRCCS, 21049 Tradate, Italy; dina.visca@icsmaugeri.it (D.V.); antonio.spanevello@icsmaugeri.it (A.S.)

**Keywords:** asthma, biologics, chronic rhinosinusitis with nasal polyps, dupilumab, lung function, real-life

## Abstract

Dupilumab is currently approved for the treatment of Type 2 severe asthma and chronic rhinosinusitis with nasal polyps (CRSwNP). Few studies have specifically reported on dupilumab efficacy on asthma outcomes as a primary objective in a real-life setting, in patients with and without CRSwNP. Our study aimed to explore the efficacy of dupilumab on functional, inflammatory, and patient-reported outcomes in asthma patients across different disease phenotypes and severity, including mild-to-moderate asthma coexisting with CRSwNP. Data from 3, 6, and 12 months follow-up were analyzed. Asthma (FEV1%, Tiffeneau%, ACT, FeNO, oral steroid use, exacerbation rate, and blood eosinophilia) and polyposis (SNOT22, VAS, NPS) outcomes showed a rapid (3 months) and sustained (6 and 12 months) significant change from baseline, despite most of the patients achieving oral steroid withdrawal. According to the sensitivity analysis, the improvement was not conditioned by either the presence of polyposis or severity of asthma at baseline. Of note, even in the case of milder asthma forms, a significant further improvement was recorded during dupilumab treatment course. Our report provides short-, medium-, and long-term follow-up data on asthma outcomes across different diseases phenotypes and severity, contributing to the real-world evidence related to dupilumab efficacy on upper and lower airways T2 inflammation.

## 1. Introduction

Dupilumab is a fully human IgG monoclonal antibody inhibiting Type 2 inflammation through a selective blocking action on IL-4 and IL-13 signaling [1]. In detail, by binding the IL-4 receptor α-subunit shared with the IL-13 receptor, dupilumab interferes with the inflammatory cascade mediated by both the cytokines. Within the frame of type 2 inflammation, IL-4 and IL-13, released by all leukocyte populations, act as pivotal mediators in the immunological response underlying clinical, functional, and structural manifestations, including airway remodeling, in asthma and chronic rhinosinusitis with nasal polyps (CRSwNP). In fact, they stimulate Th2 cells differentiation, IgE production, eosinophil, basophil and mast cells activation, smooth muscle proliferation, fibrosis, and mucus production [2]. The IL-4R-driven axis represents a crucial target for precision medicine therapies with the aim to limit allergic and T2 inflammation and intervene in disease chronicity both in adults and in children.

Following the efficacy and safety evidence provided by several trials, dupilumab is currently licensed and marketed for the treatment of Type 2 severe asthma [3,4] and moderate-to-severe atopic dermatitis, in adult and children, from the age of 6 years [5,6], as well as for CRSwNP from the age of 18 years [7,8]. Overall, randomized controlled trials showed a 50–60% reduction in asthma exacerbation, 70% glucocorticoid dose reduction, and significative improvement in lung function in severe asthma patients treated with dupilumab when compared to placebo [3,4].

So far, few studies have specifically reported on dupilumab efficacy on asthma outcomes in a real-life setting, most of them including a small sample size and a limited follow-up time frame [9,10,11,12,13,14,15,16,17,18]. In addition, despite the fact that the coexistence of CRSwNP and asthma does represent a well-known factor for worse and more difficult asthma control [19], whether such comorbidity limits the effect of dupilumab on asthma outcomes in treated patients has been poorly investigated in real life [14,15]. Furthermore, the available data are mostly limited to severe asthma patients [12]. 

Our study aimed to explore the efficacy of dupilumab on functional, inflammatory, and patient-reported outcomes in asthma patients across different disease phenotypes and severity. We analyzed patients with severe asthma with or without nasal polyps, patients with CRSwNP with coexisting mild-to-moderate asthma, and patients suffering from Samter’s triad.

## 2. Materials and Methods

A prospective multicenter observational study involving thirteen referral centers for severe asthma across Italy (Bari, Brescia, Cagliari, Catania, Genoa, Milan, Modena, Naples, Padua, Siena, Turin, Varese, Verona) was performed. Consenting patients referring to the participating centers between March and December 2022 and prescribed with dupilumab at the approved dose of 300 mg/2 weeks subcutaneously for severe asthma and/or chronic rhinosinusitis with nasal polyps (CRSwNP) were enrolled. More in detail, the inclusion criteria were the following: A confirmed diagnosis of severe asthma according to the ERS/ATS definition [20]; CRSwNP according to EUFOREA/EPOS definition [21] with severe or non-severe asthma, in the last case according to GINA definition [22]; eligibility for dupilumab treatment, according to the prescription requirements established by the European Regulatory Agency [23]. Prior treatment with other biologics for the same indications was considered an exclusion criterion. Furthermore, in accordance with the study aim, patients with CRSwNP only and without asthma were also excluded.

Data from 3, 6, and 12 months follow-up were analyzed. More in detail, changes over time in lung function (spirometry assessment, including FEV1 and Tiffenau index), inflammation biomarkers (exhaled nitric oxide—FeNO, blood eosinophils), clinical parameters (oral steroid use; annual asthma exacerbation rate, defined as the need for an oral steroid trial or an increase in the daily dose), and patients reported outcomes (asthma control test—ACT) were assessed. For CRSwNP patients, sino-nasal outcome test (SNOT) 22 score, nasal polyp score (NPS), and visual analogic scale (VAS) related to smell were also considered. The evaluation was performed according to current clinical practice standards.

Dupilumab-related adverse event potentially occurring over the study timeframe were also reported and evaluated. 

### Statistical Analysis

A descriptive statistic was first performed to explore patients’ characteristics at baseline. Percentages and frequency rates were used for categorical variables and medians with interquartile range for continuous ones. Differences in sample distribution at baseline were assessed through chi-squared and Fisher’s exact test or Mann–Whitney U non-parametric test, as appropriate. 

Asthma (Fev1%, Tiffeneau%, ACT, FeNO, OCS use probability, eosinophilia) and polyposis (SNOT22, VAS, NPS) outcome measurements were modeled using linear mixed-effect (LMER) and generalized linear mixed-effects (GLMER) models for longitudinal data and subject-specific random effects to estimate evolutions of the target outcomes for patients over the course of the follow-up and to compare predicted trajectories adjusted for independent variables (sex, age, type of diagnosis).

Statistical analyses were performed in R v4.3, using additional packages lme4 [24], ggplot2 [25], and ggeffects [26] to produce the growth charts. A *p*-value < 0.05 was considered significant.

## 3. Results

### 3.1. Sample Characteristics

In the study period, 195 patients were included in the analysis. Table 1 summarizes the main features of the study population. The median age was 55 years (IQR 45–64) and 102 (52.3%) were females with no differences by age compared to male (*p* = 0.091). Most patients were diagnosed with severe asthma and polyposis (*n* = 90, 46.1%), 39 (20.0%) had only severe asthma, 37 (19.0%) and 29 (14.9%) had mild-to-moderate asthma and polyposis and Samter’s Triad, respectively. Patients with mild-to-moderate asthma and polyposis had a median age of 47 years (IQR 42–61) and were younger compared to those with severe asthma and polyposis (58 years, IQR 49–64). There were no differences in diagnosis type based on sex (*p* = 0.118).

At least one chronic comorbidity was found in 85 (43.6%) patients, with a median number of 2 comorbidity (IQR 1–3) per patient. Those with comorbidity were older than the others (*p* < 0.001), without differences based on sex (*p* = 0.392). Most frequent comorbidities were GERD (*n* = 29, 34.1%), hypertension (*n* = 25, 29.4%), and osteoporosis (*n* = 18, 21.2%). There were no differences in comorbidities between asthma severity and phenotype (*p* = 0.103). Atopy was recorded in 105 patients (53.8%).

### 3.2. Asthma Outcomes

The trend of asthma outcomes over the study timeframe are summarized by Figure 1 and Figure 2. Appendix A provide individual data plotting.

#### 3.2.1. FEV1 (Figure 1)

The estimated FEV1 percentage at baseline was 83.7 [0.95 CI 80.3–87.0] and increased to 92.4 [0.95 CI 88.4–96.4] after three months (*p* < 0.001). Then the value remained significantly higher after 6 (91.2 [0.95 CI 87.5–94.9], *p* < 0.001) and 12 (95.0 [0.95 CI 90.7–99.2], *p* < 0.001) months of treatment compared to baseline without further increases compared to the third month value (*p* = 0.850, *p* = 0.749). FEV1 was lower as age increased (beta = −0.21, SE = 0.10, *p =* 0.034) and in those with asthma compared to patients with asthma plus polyps (beta = 8.05, SE = 3.15, *p =* 0.011). No differences were found on the FEV1 at any time-point compared to t0 based on FeNO at baseline. In the sensitivity analysis, no differences were found in the FEV1 trend based on neither the presence of polyposis nor severity of asthma.

#### 3.2.2. Tiffeneau Index (Figure 1)

The estimated Tiffeneau percentage at baseline was 74.8 [0.95 CI 72.4–77.4] and increased to 77.9 [0.95 CI 74.1–80.8] after three months of treatment (*p* = 0.004). Then the value remained significantly higher after 6 (79.0 [0.95 CI 76.4–81.7], *p* < 0.001) and 12 (79.3 [0.95 CI 76.3–82.3], *p* = 0.004) months of treatment compared to baseline without further increases compared to the third month value (*p =* 0.831, *p* = 0.997). Tiffeneau was generally lower as age increased (beta = −0.27, SE = 0.07, *p* < 0.001) without differences based on diagnosis or sex (*p =* 0.295, *p =* 0.269). In the sensitivity analysis, no differences were found in the Tiffeneau trend based on neither the presence of polyposis nor severity of asthma.

#### 3.2.3. ACT (Figure 1)

The estimated ACT score at baseline was 16.7 [0.95 CI 15.9–17.6] and increased to 20.1 [0.95 CI 19.1–21.1] after three months of treatment (*p* = <0.001). Then the value remained significantly higher after 6 (21.4 [0.95 CI 20.5–22.4], *p* < 0.001) and 12 (21.5 [0.95 CI 20.2–22.7], *p <* 0.001) months of treatment compared to baseline without further increases compared to the third month value (*p* = 0.239, *p =* 0.998). ACT score was generally lower in those with asthma compared to patients with asthma plus polyps (beta = 2.73, SE = 0.69, *p* < 0.001). In the sensitivity analysis, no differences were found in the ACT score trend based on neither presence of polyposis nor severity of asthma.

**Figure 1 biomedicines-12-00390-f001:**
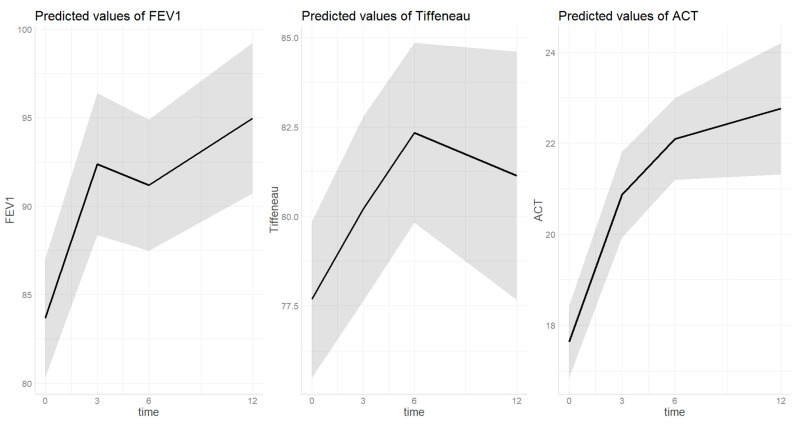
Trend of lung function parameters over dupilumab treatment. Time on X-axis is expressed in months. The grey area indicates the 0.95 confidential interval

#### 3.2.4. OCS (Figure 2)

The estimated probability of being on treatment with OCS at baseline was 0.812 [0.95 CI 0.709–0.885] and decreased to 0.113 [0.95 CI 0.053–0.226] after three months of treatment (*p* ≤ 0.001). Then the value remained significantly lower after six (0.116 [0.95 CI 0.060–0.214], *p <* 0.001) and twelve (0.093 [0.95 CI 0.041–0.197], *p <* 0.001) months of treatment with dupilumab compared to baseline without further deceased compared to the third month value (*p =* 0.239, *p* = 0.998). There were no differences in OCS treatment probability based on sex (*p =* 0.654), age (*p* = 0.104) or type of diagnosis (*p* = 0.054). In the sensitivity analysis, the probability of being on treatment with OCS after 3 months of treatment was significantly lower in those with mild-to-moderate asthma (0.013 [0.95 CI 0.001–0.097]) than those with severe (0.189 [0.95 CI 0.089–0.357], *p =* 0.033). After 6 (*p =* 0.071) and 12 (*p =* 0.082) months of treatment this difference was no longer found. No difference was found in OCS probability trend based on presence of polyposis.

#### 3.2.5. FeNO (Figure 2)

The estimated FeNO value at baseline was 46.8 ppb [0.95 CI 41.5–52.2] and decreased to 26.4 ppb [0.95 CI 19.2–33.7] after three months of treatment (*p* < 0.001). Then the value remained significantly lower after six (27.0 [0.95 CI 20.7–32.2], *p* < 0.001) and twelve (18.8 [0.95 CI 11.2–26.3], *p* < 0.001) months of treatment with dupilumab compared to baseline without further deceased compared to the third month value (*p* = 0.361, *p* = 0.295). There were no differences in FeNO value based on sex (*p* = 0.804), age (*p* = 0.659), or type of diagnosis (*p* = 0.067). In the sensitivity analysis, no differences were found in the FeNO trend based on the presence of polyposis or severity of asthma.

#### 3.2.6. Blood Eosinophils (Figure 2)

The estimated eosinophils count value increased from 409.9 [0.95 CI 314.2–505.7] at baseline to 1128.2 [0.95 CI 1003.2–1253.2] after three months of therapy (*p <* 0.001). This value remained significantly higher compared to baseline at 6th (993.5 0.95 CI 835.2–1151.7, *p <* 0.001) and 12th (836.3 [0.95 CI 639.2–1033.4], *p* < 0.001) month of follow up. No differences were found based on sex (*p* = 0.992), age (*p* = 0.268), or type of diagnosis (*p* = 0.448). In the sensitivity analysis, the estimated eosinophilia value after 3 months of treatment was significantly higher in those with severe asthma (1225.6 [0.95 CI 1075.9–1375.2]) than those with mild-to-moderate (1082.4 [0.95 CI 921.2–1243.6], *p* = 0.037). After 6 (*p* = 0.085) and 12 (*p =* 0.996) months of treatment this difference was no longer found. No difference was found in OCS probability trend based on presence of polyposis.

#### 3.2.7. Asthma Exacerbations (Figure 2)

The estimated number of asthma exacerbations decreased from 2.3 (0.95 CI 2.0–2.6) at baseline to 0.2 (0.95 CI 0.0–0.7, *p <* 0.001) and 0.1 (0.95 CI 0.0–0.6, *p* < 0.001) after 6 and 12 months of therapy. There were no differences based on age (*p* = 0.389), sex (*p* = 0.149), or diagnosis (*p* = 0.908). In the sensitivity analysis, no differences were found in the asthma exacerbations trend based on presence of polyposis or severity of asthma.

**Figure 2 biomedicines-12-00390-f002:**
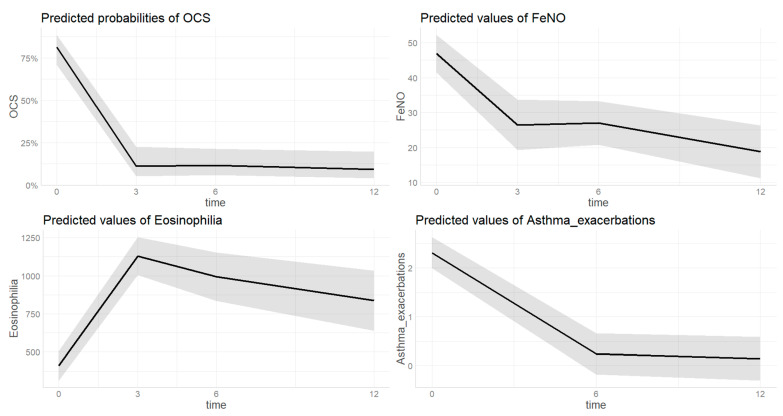
Change of different asthma-related outcomes during dupilumab treatment; OCS = oral corticosteroids. Time on X-axis is expressed in months. The grey area indicates the 0.95 confidential interval.

### 3.3. Polyposis Outcomes

Considering the sample of patients with asthma plus polyposis (*n* = 156), the following outcomes were analyzed. The trend of polyposis outcomes over the study timeframe are summarized by Figure 3. Appendix A provides individual data plotting. 

#### 3.3.1. SNOT22 (Figure 3)

The estimated SNOT22 score at baseline was 59.4 [0.95 CI 55.9–42.8] and decreased to 27.4 [0.95 CI 23.5–31.3] after three months of treatment (*p* = <0.001). Then a further decrease was observed at the 6th month of treatment with dupilumab (20.8 [0.95 CI 17.0–24.5], *p* = 0.031). After twelve months of treatment, the score remained stable and significantly lower (18.8 [0.95 CI 13.5–24.1]) compared to both baseline (*p* < 0.001) and third month (*p* = 0.025). SNOT22 score was not significantly different based on sex (*p =* 0.303), age (*p =* 0.113), or presence of atopy (*p =* 0.095).

#### 3.3.2. VAS (Figure 3)

The estimated VAS score at baseline was 11.1 [0.95 CI 8.9–13.3] and decreased to 5.9 [0.95 CI 3.6–8.2] after three months of treatment (*p ≤* 0.001). Then the value remained significantly lower after six (5.3 [0.95 CI 3.0–7.6], *p <* 0.001) and twelve (5.4 [0.95 CI 3.1–7.8], *p* < 0.001) months of treatment with dupilumab compared to baseline without further decease compared to the third month value (*p =* 0.885, *p =* 0.998). VAS score was not significantly different based on sex (*p =* 0.054), age (*p =* 0.768), or presence of atopy (*p =* 0.531).

#### 3.3.3. NPS (Figure 3)

The estimated VAS score at baseline was 5.4 [0.95 CI 5.0–5.8] and decreased to 2.4 [0.95 CI 1.9–2.8] after three months of treatment (*p ≤* 0.001) and remained stable at the sixth month (2.0[1.6–2.5], *p* = 0.719). After twelve months of treatment, a further decrease compared to the third month was observed, reaching 1.4 (0.95 CI 0.8–2.0, *p* = 0.037). NPS score was not significantly different based on sex (*p* = 0.0.069), age (*p* = 0.368), or presence of atopy (*p =* 0.297).

**Figure 3 biomedicines-12-00390-f003:**
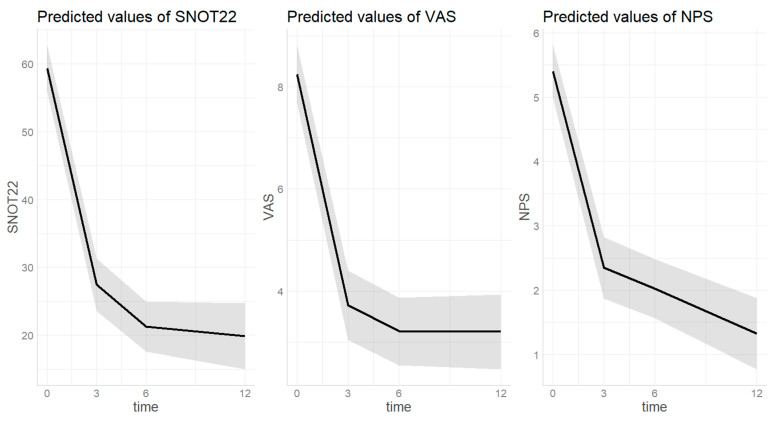
Trend of nasal-polyps-related outcomes over dupilumab treatment. Time on X-axis is expressed in months. SNOT22 = sinonasal outcome test; VAS = visual analogic scale; NPS = nasal polyp score. The grey area indicates the 0.95 confidential interval.

### 3.4. Safety

No severe adverse events were reported over the study timeframe. In 10 cases, clinical manifestations not previously reported by the patients, not related to the conditions treated with dupilumab, and, thus, reported as adverse events were observed. More in detail, six patients presented myalgias, in two cases conjunctivitis occurred, and headache and paresthesia was observed in one case each. 

## 4. Discussion

Our real-life study explored the efficacy of dupilumab when prescribed to patients with severe asthma with or without CRSwNP and to patients with CRSwNP and coexisting mild-to-moderate asthma. When looking at both objective parameters and patient-reported outcomes, our findings confirm a rapid and maintained effect of dupilumab over 12 months follow up, including a dramatic steroid-sparing effect. Coexisting CRSwNP did not affect the dupilumab efficacy profile in treated patients.

A limited number of studies in a real-life setting have focused on severe asthma outcomes as the primary objective, most of them following up with patients for 6 months or less [10,11,13,14,16]. or including a small sample size (64 patients in a French cohort [9], 18 patients in an Italian study [18]). Our results confirm the observations on a large cohort (148 patients) from the Dutch real-word registry [17], describing a significant improvement in FEV1, FeNO, blood eosinophils count, asthma control questionnaire, and OCS dependency at the 6 months follow-up, together with the reduction in annual exacerbation rate as well as the maintenance of the previous achievements at the 12 months assessment. In our study, we were able to demonstrate an early dupilumab effect on all the functional, inflammatory, and patient-reported outcomes at the 3 months follow-up, subsequently sustained for the whole year of observation (Figure 1 and Figure 2). Those findings are even more relevant when considering the very fast OCS tapering or withdrawal for most patients, suggesting that the observed rapid and maintained improvement can be related uniquely to the dupilumab effect. Of note, when compared to the randomized clinical trials, our data show an even greater steroid-sparing effect of dupilumab: in fact, after 12 months of treatment, more than 85% of the patients were able to discontinue OCS, while in the TRAVERSE study around 60% of patients achieved a significant steroid dependence reduction [27].

When looking at CRSwNP and Samter’s triad patients’ subgroup, according to the sensitivity analysis of our data, the coexistence of upper and lower airways disease did not significantly impact on dupilumab efficacy profile, in terms of both rapidity and entity, including all the functional, inflammatory, and patient-reported outcomes. That observation is even more relevant when considering the well-known association between nasal polyps and a more difficult-to-treat asthma form; in fact, patients with asthma and comorbid nasal polyposis have higher airway inflammation, lower disease control, higher exacerbation rate, and need higher OCS doses [19,28]. Furthermore, the evidence that dupilumab efficacy on asthma control is independent by the presence of the upper airways comorbidity might be helpful in the correct positioning of the drug under a personalized treatment approach perspective. 

In addition, objective and patient-reported outcomes related to CRSwNP followed the trend of asthma outcomes in our population (Figure 3), demonstrating a rapid and significant direct effect of dupilumab on upper airways. Our results are in line with a recently published real-life study [14], although limited to a 6 months follow-up, and with the work by Al-Ahmad et al. [15]. The authors explored as a secondary objective the effect of dupilumab on asthma symptom control. However, differently from our study, the only considered asthma outcome was ACT, the upper airways evaluation did not consider any objective parameters, and the treatment duration was heterogeneous within the study population.

We also had the chance to explore clinical functional and inflammatory asthma outcomes in non-severe forms of the disease in patients prescribed with dupilumab for CRSwNP and suffering from coexisting mild-to-moderate asthma. The sensitivity analysis demonstrates that the change in asthma outcomes overtime does not significantly differ according to asthma severity at baseline. It suggests that even in the case of milder asthma forms, a further improvement can be achieved during dupilumab treatment course. After all, it is known that even in the presence of normal lung function parameters, significant post-bronchodilator reversibility can be detected in asthma patients [29] and that persistent bronchodilator response in treated asthma patients correlates with lower disease control [30]. Similar findings were reported by Minagawa et al. [12], who conducted a study on 62 CRSwNP patients, 50 of them with comorbid mild-to-moderate asthma and 12 with severe asthma. When comparing lung function, FeNO, and ACT by asthma severity, they did not find any statistically significant difference. However, differently from our analysis, the proposed comparison refers to the 3 months follow-up.

Our findings should be interpreted in the light of some study limitations, including the lack of randomization and the different sample size of the analyzed subgroups. However, as an element of novelty, our report provides short-, medium-, and long-term follow up data on asthma functional, inflammatory, and patient-reported outcomes across different diseases phenotypes and severity, contributing to the real-world evidence related to dupilumab efficacy on upper and lower airways T2 inflammation.

## Figures and Tables

**Table 1 biomedicines-12-00390-t001:** Overview of the study population’s characteristics.

	Severe Asthma with Nasal Polyps (*n* = 90)	Severe Asthma (*n* = 39)	Mild Asthma with Nasal Polyps (*n* = 37)	Samter–Widal (*n* = 29)	Overall (*n* = 195)
Sex					
Female	42 (46.7%)	20 (51.3%)	19 (51.4%)	21 (72.4%)	102 (52.3%)
Male	48 (53.3%)	19 (48.7%)	18 (48.6%)	8 (27.6%)	93 (47.7%)
Age					
Median (IQR)	57.5 (49.3–6.0)	54.0 (43.0–62.0)	47.2 (42.0–61.3)	56.0 (49.0–64.1)	55.0 (45.0–64.0)
Atopy					
No	49 (54.4%)	9 (23.1%)	21 (56.8%)	11 (37.9%)	90 (46.2%)
Yes	41 (45.6%)	30 (76.9%)	16 (43.2%)	18 (62.1%)	105 (53.8%)
N° comorbidities					
0	48 (53.3%)	18 (46.2%)	27 (73.0%)	17 (58.6%)	110 (56.4%)
1	17 (18.9%)	9 (23.1%)	7 (18.9%)	9 (31.0%)	42 (21.5%)
2	10 (11.1%)	4 (10.3%)	1 (2.7%)	0 (0.0%)	15 (7.7%)
3	9 (10.0%)	4 (10.3%)	0 (0.0%)	2 (6.9%)	15 (7.7%)
≥4	6 (6.7%)	4 (10.3%)	2 (5.4%)	1 (3.4%)	13 (6.7%)
OCS at baseline					
None	23 (25.6%)	10 (25.6%)	13 (35.1%)	9 (31.0%)	55 (28.2%)
On demand	26 (28.9%)	16 (41.0%)	22 (59.5%)	17 (58.6%)	81 (41.5%)
Low dose	12 (13.3%)	7 (17.9%)	0 (0.0%)	2 (6.9%)	21 (10.8%)
Middle dose	10 (11.1%)	4 (10.3%)	1 (2.7%)	0 (0.0%)	15 (7.7%)
High dose	10 (11.1%)	2 (5.1%)	1 (2.7%)	1 (3.4%)	14 (7.2%)

OCS = oral corticosteroids; low dose ≤ 5 mg/day; medium dose = 5–10 mg/day; high dose ≥ 10 mg prednisolone or equivalent.

## Data Availability

Data are available upon request.

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
