# Peer review of "Dupilumab Efficacy on Asthma Functional, Inflammatory, and Patient-Reported Outcomes across Different Disease Phenotypes and Severity: A Real-Life Perspective"

_biomedicines, 2024, doi:10.3390/biomedicines12020390_

Round 1

Reviewer 1 Report

Comments and Suggestions for Authors

This article examines the effectiveness of dupilumab, (a fully humanized IgG monoclonal antibody that targets IL-4 and IL-13 signaling), in patients with various asthma phenotypes and severities, it also evaluates its performance in daily life. The article begins by explaining the mechanism of action of dupilumab and its approved use in conditions like severe asthma and atopic dermatitis. It highlights the limitations of previous studies, such as small sample sizes and short follow-up periods, and emphasizes the need for further research on the efficacy of dupilumab in patients with different asthma phenotypes and severities, as well as in patients with or without chronic rhinosinusitis with nasal polyps (CRSwNP). However, there are still some minor modifications needed to address certain shortcomings.

1. In the introduction, the article briefly mentions IL-4 and IL-13 as two important cytokines in allergic diseases like asthma and chronic rhinosinusitis with nasal polyps (CRSwNP). However, it lacks specific information about the pathogenesis of these cytokines, which could be added for a better understanding.

2. The transitions in the preface are not smooth, as seen in the sentence 'the potential impact of CRSwNP on asthma outcomes in treated patients has been poorly investigatedand a small amount of data refers non-severe asthma forms.' This sentence lacks logical coherence and clarity. 

3. The article lacks specific results or data regarding the effectiveness of dupilumab in treating asthma. Consequently, evaluating the drug's efficacy across various patient phenotypes and levels of severity becomes a challenge.

4. The article fails to mention any side effects or adverse reactions associated with dupilumab. This omission prevents an accurate assessment of the drug's efficacy and safety.

5. The article fails to present any comparative data between dupilumab and alternative asthma treatments. This absence of data renders it challenging to understand how the drug stacks up in terms of its advantages and disadvantages compared to other existing treatment options.

6. The full text is centered around the efficacy of dupilumab in the treatment of asthma and with or without comorbid CRSwNP, but it lacks a description of the significance of the specific clinical manifestations regarding the combination of these two conditions.

7. The Materials and Methods section is missing specific experimental procedures and conditions. Only the results are analyzed, making it difficult to assess the feasibility of the method. 

8. In the Discussion section, it is suggested to add literature support. For example, 'nasal polyps are associated with a more difficult to treat asthma form.

9. The article should include a complete list of abbreviations. For example, the abbreviation IQR used in the experimental results should be added.

10.'Pin the literature needs to be italicized, it is suggested to change.

Author Response

This article examines the effectiveness of dupilumab, (a fully humanized IgG monoclonal antibody that targets IL-4 and IL-13 signaling), in patients with various asthma phenotypes and severities, it also evaluates its performance in daily life. The article begins by explaining the mechanism of action of dupilumab and its approved use in conditions like severe asthma and atopic dermatitis. It highlights the limitations of previous studies, such as small sample sizes and short follow-up periods, and emphasizes the need for further research on the efficacy of dupilumab in patients with different asthma phenotypes and severities, as well as in patients with or without chronic rhinosinusitis with nasal polyps (CRSwNP). However, there are still some minor modifications needed to address certain shortcomings.

  1. In the introduction, the article briefly mentions IL-4 and IL-13 as two important cytokines in allergic diseases like asthma and chronic rhinosinusitis with nasal polyps (CRSwNP). However, it lacks specific information about the pathogenesis of these cytokines, which could be added for a better understanding.

R: Further details about the pathobiological relevance of the mentioned cytokines have been included.

  1. The transitions in the preface are not smooth, as seen in the sentence 'the potential impact of CRSwNP on asthma outcomes in treated patients has been poorly investigatedand a small amount of data refers non-severe asthma forms.' This sentence lacks logical coherence and clarity.

R: The text has been modified in order to clarify the flow.

  1. The article lacks specific results or data regarding the effectiveness of dupilumab in treating asthma. Consequently, evaluating the drug's efficacy across various patient phenotypes and levels of severity becomes a challenge.

R: A brief summary related to dupilumab efficacy evidence from the published trials has been included in the introduction as well as in the discussion.

  1. The article fails to mention any side effects or adverse reactions associated with dupilumab. This omission prevents an accurate assessment of the drug's efficacy and safety.

R: Thank you for raising this relevant point. Information has been included.

  1. The article fails to present any comparative data between dupilumab and alternative asthma treatments. This absence of data renders it challenging to understand how the drug stacks up in terms of its advantages and disadvantages compared to other existing treatment options.

R: Thank you for the comment. A comparative analysis between dupilumab and other biologic treatments for asthma would present some methodological bias. In fact dupilumab, as well as the other biologic therapies currently licensed for the disease reflects under both clinical and regulatory perspectives a precision medicine approach. In other words, patients who are eligible to dupilumab treatment are characterized by a different clinical and pathobiological background when compared to potential candidates to other target treatments. For that reason studies on biologic therapies for asthma are used to compare the same patients before and after the treatment.

  1. The full text is centered around the efficacy of dupilumab in the treatment of asthma and with or without comorbid CRSwNP, but it lacks a description of the significance of the specific clinical manifestations regarding the combination of these two conditions.

R: Additional information has been included in the introduction and discussion sessions

  1. The Materials and Methods section is missing specific experimental procedures and conditions. Only the results are analyzed, making it difficult to assess the feasibility of the method.

R: Materials and methods section was better detailed.

  1. In the Discussion section, it is suggested to add literature support. For example, 'nasal polyps are associated with a more difficult to treat asthma form.'

R: Additional references have been provided.

  1. The article should include a complete list of abbreviations. For example, the abbreviation IQR used in the experimental results should be added.

R: A full list of abbreviations has been included.

10.'P' in the literature needs to be italicized, it is suggested to change.

R: the text has been modified accordingly

Reviewer 2 Report

Comments and Suggestions for Authors

The present manuscript shows the results od a study of the efficacy of a monoclonal antibody (Dupilumab) in asthma.

I its of great interest to have information about the benefices of the new drugs that have been introduced in the therapy in recent times.

The authors show data of an important group of patients, measuring different aspects related with the disease.

The work has been done in a very organized way and the results are easy to interpret.

Something to improve:

-          The authors do no include in the manuscript the dosage and way of administration of the drug

-          In the figures the axis are difficult to see. I recommend to increase the size of the font used and also the colour

-          Even though it is not the objective of the present work, I consider that it would be important to know if the patients of the study had frequent or serious adverse reactions with Dupilumab.

Considering the benefits that can bring the drug in the treatment of asthma, this is an important point to consider, as well as the economic factor

Author Response

The present manuscript shows the results od a study of the efficacy of a monoclonal antibody (Dupilumab) in asthma.

I its of great interest to have information about the benefices of the new drugs that have been introduced in the therapy in recent times.

The authors show data of an important group of patients, measuring different aspects related with the disease.

The work has been done in a very organized way and the results are easy to interpret.

Something to improve:

-         The authors do no include in the manuscript the dosage and way of administration of the drug

R: The requested information has been included

-         In the figures the axis are difficult to see. I recommend to increase the size of the font used and also the colour

R: Thank you for the comment. The figures have been revised accordingly

-         Even though it is not the objective of the present work, I consider that it would be important to know if the patients of the study had frequent or serious adverse reactions with Dupilumab. Considering the benefits that can bring the drug in the treatment of asthma, this is an important point to consider, as well as the economic factor

  1. Thank you for highlighting that relevant point. The information has been included.